# The Effect of Probiotics on Gut Microbiota Modulation and Its Role in Mitigating Diabetes-Induced Hepatic Damage in Wistar Rats

**DOI:** 10.3390/biology14040323

**Published:** 2025-03-22

**Authors:** Alaa Talal Qumsani

**Affiliations:** Biology Department, Al-Jumum University College, Umm Al-Qura University, Makkah 24382, Saudi Arabia; atqumsani@uqu.edu.sa

**Keywords:** gut microbiota modulation, probiotic supplementation, metformin, diabetes-induced hepatic dysfunction, *Bifidobacterium bifidum*, liver function preservation, oxidative stress, hepatoprotection

## Abstract

The gut microbiota plays a crucial role in maintaining overall health and preventing various diseases, including its protective effects against diabetes-induced liver injury. This study investigates the potential of *Bifidobacterium bifidum* as a probiotic intervention to support liver function under diabetic conditions. Four experimental groups were included: a healthy control group, a probiotic-only group, a diabetic group, and a diabetic group treated with probiotics. Probiotic supplementation was initiated 2 weeks before diabetes induction and continued throughout the study. The results demonstrated that probiotic administration significantly improved gut microbiota composition, enhanced glycemic control, and increased insulin sensitivity in diabetic rats. Probiotic-treated diabetic rats exhibited improved microbial diversity, enhanced gut health markers, and reduced oxidative stress. Liver histopathology revealed preserved structural integrity, reduced inflammation, and decreased DNA damage. These findings suggest that targeting the gut microbiota with probiotics may be a promising therapeutic approach for managing diabetes-associated liver complications. Further studies are required to optimize probiotic interventions and assess their long-term efficacy in diabetes management.

## 1. Introduction

Diabetes mellitus (DM) is a chronic metabolic disorder characterized by persistent hyperglycemia resulting from impaired insulin production, secretion, or function. The global prevalence of DM is rising at an alarming rate, with projections estimating that over 700 million individuals will be affected by 2045 [1]. Among its many complications, liver disease, particularly non-alcoholic fatty liver disease (NAFLD) has emerged as a major concern, affecting up to 70% of individuals with type 2 diabetes (T2D) [2]. The relationship between DM and liver dysfunction is bidirectional: hepatic insulin resistance contributes to glucose dysregulation, while metabolic inflammation further exacerbates hepatic injury.

Recent research has increasingly highlighted the critical role of the gut microbiota in modulating liver function, maintaining metabolic homeostasis, and regulating immune responses [3]. The gut microbiota, a complex ecosystem of trillions of microorganisms, is essential for numerous physiological functions, including digestion, metabolism, and immune regulation [4]. Disruptions in microbial composition—referred to as dysbiosis—have been strongly linked to metabolic disorders, systemic inflammation, and liver dysfunction [5]. Emerging evidence indicates that gut dysbiosis may contribute to NAFLD and diabetes-induced hepatic injury through mechanisms such as increased gut permeability, endotoxemia, and altered bile acid metabolism [6]. Short-chain fatty acids (SCFAs), especially butyrate, propionate, and acetate, are key microbial metabolites that regulate glucose metabolism and hepatic function by modulating inflammatory signaling, enhancing intestinal barrier integrity, and protecting against hepatic fibrosis via inhibition of the NF-κB pathway [7].

Individuals with T2D often exhibit a marked reduction in beneficial bacterial strains—such as *Akkermansia muciniphila* and *Faecalibacterium prausnitzii*—which are essential for maintaining gut homeostasis, controlling systemic inflammation, and supporting liver health [8]. The loss of these beneficial microbes is associated with increased intestinal permeability, which facilitates the translocation of bacterial endotoxins (e.g., lipopolysaccharides, LPS) into the bloodstream. This translocation triggers inflammatory cascades that exacerbate insulin resistance and accelerate hepatic fibrosis [9]. Additionally, dysbiosis can disrupt bile acid metabolism, leading to an accumulation of toxic secondary bile acids that impair liver function and contribute to overall metabolic dysfunction [10].

Given the substantial impact of gut dysbiosis on liver disease, microbiota-targeted therapies have garnered increasing attention as potential interventions for diabetes-related hepatic injury. Probiotic supplementation—particularly with strains from the *Bifidobacterium bifidum* and *Akkermansia* genera—has shown promising benefits in restoring microbial diversity, reducing systemic inflammation, and improving hepatic outcomes [11]. Probiotics have been demonstrated to enhance intestinal barrier integrity, reduce oxidative stress markers, and regulate lipid metabolism, thereby mitigating hepatic steatosis and fibrosis [12]. Notably, recent studies have confirmed that *Bifidobacterium bifidum* supplementation enhances hepatic insulin sensitivity, reduces liver fat accumulation, and modulates gut-derived inflammatory responses in diabetic models [13].

Probiotics—especially those belonging to the *Bifidobacterium* and *Lactobacillus* genera—play a significant role in sustaining gut microbial balance and enhancing metabolic health. They boost the production of short-chain fatty acids (SCFAs), notably butyrate, which helps regulate hepatic lipid metabolism and improve insulin sensitivity. Additionally, probiotics fortify gut barrier integrity, reducing lipopolysaccharide (LPS)-induced inflammation and systemic endotoxemia. Clinical and experimental evidence underscores how probiotic interventions can mitigate diabetes-related liver injury by restoring microbial diversity and diminishing hepatic oxidative stress [14]. Recent research also indicates that combining probiotics with functional dietary interventions can further optimize gut microbiota modulation and improve both metabolic and hepatic outcomes [15].

### Study Objective

This study aims to evaluate the therapeutic potential of *Bifidobacterium bifidum* in mitigating diabetes-induced hepatic injury by modulating gut microbiota composition. By assessing changes in microbial diversity, inflammatory markers, and liver function—and by comparing the effects of probiotic supplementation alone, metformin treatment, and their combination—this research seeks to elucidate the intricate relationship between a balanced gut microbiota and liver health. The findings of this study will contribute to the growing body of evidence supporting gut microbiota-targeted therapies as a promising strategy for managing diabetes-related liver complications.

## 2. Methodology

### 2.1. Experimental Design and Animal Model

The study investigated the hepatoprotective effects of *Bifidobacterium bifidum* ATCC 29521 on diabetes-induced liver damage in Wistar rats. A total of 60 male rats (aged six months, 180–200 g) were randomly divided into six experimental groups (*n* = 10 per group):

Control group (C): Healthy rats receiving no treatment, serving as a baseline for physiological and metabolic parameters.

Probiotic-only group (B): Healthy rats receiving daily supplementation with *Bifidobacterium bifidum* at a dose of 1 × 10^9^ CFU/kg/day to evaluate any direct effects of the probiotic on normal liver physiology.

Diabetic group (D): Rats in which diabetes was induced using a single intraperitoneal injection of streptozotocin (STZ, 50 mg/kg; Sigma-Aldrich, St. Louis, MO, USA) after a 12 h fast, without any probiotic supplementation.

Diabetic + probiotic group (D+B): Diabetic rats treated with *Bifidobacterium bifidum* at a dose of 1 × 10^9^ CFU/kg/day to determine the probiotic’s therapeutic impact on liver injury associated with diabetes.

Diabetic + metformin group (D+M): Diabetic rats treated with metformin at a dose of 200 mg/kg/day, serving as a standard treatment for hepatoprotection.

Diabetic + probiotic + metformin group (D+B+M): Diabetic rats treated with a combination of *Bifidobacterium bifidum* (1 × 10^9^ CFU/kg/day) and metformin (200 mg/kg/day), allowing for the investigation of potential synergistic effects in mitigating diabetes-induced hepatic damage.

Rats were housed in a temperature-controlled environment (22 ± 2 °C, 50–60% relative humidity, 12-h light/dark cycle) in ventilated cages with corn cob bedding (changed twice weekly). They had free access to a standardized chow diet (Lab Diet 5001, St. Louis, MO, USA) and water ad libitum throughout the study.

### 2.2. Diabetes Induction and Probiotic Administration

Diabetes was induced after a 12 h fast using a single intraperitoneal injection of STZ (50 mg/kg; Sigma-Aldrich, St. Louis, MO, USA). Rats with fasting blood glucose (FBG) levels > 240 mg/dL on two consecutive measurements were confirmed as diabetic [16,17]. Probiotic supplementation was administered orally via gavage at a dose of 1 × 10^9^ CFU/kg body weight per day of *Bifidobacterium bifidum* ATCC 29521, prepared at a concentration of 1 × 10^9^ CFU/mL and suspended in 1 mL of sterile phosphate-buffered saline (PBS). Probiotic treatment began 2 weeks before diabetes induction and was continued daily for 12 weeks [18,19]. The sample size (*n* = 10 per group) was determined with a priori power analysis using G*Power 3.1 software, ensuring 80% power to detect significant differences at α = 0.05, with a Cohen’s d effect size of 0.8. All experimental protocols followed the AR-RIVE guidelines and were approved by the ethical committee of Umm Al-Qura University (Approval No. HAPO-02-K-012-2023-02-1157).

### 2.3. Blood Sample Collection

Weekly fasting blood samples were obtained from the distal tail vein after an overnight fast (10–12 h). At the end of the 12-week study, rats were humanely euthanized, and cardiac puncture was performed for terminal blood collection. Serum samples were obtained by allowing the blood to clot at room temperature for 30 min, followed by centrifugation (3000× *g* for 10 min). Plasma samples were collected in EDTA-coated tubes to prevent coagulation and centrifuged under the same conditions. All samples were stored at −80 °C until biochemical analysis [20].

### 2.4. Biochemical and Metabolic Assessments

#### 2.4.1. Body Weight and Fasting Blood Glucose

Animals were fasted overnight (12 h) prior to the initiation of the experiment, and diabetes was induced using a single intraperitoneal injection of streptozotocin (Model: STZ-100, Sigma-Aldrich, St. Louis, MO, USA) at a dose of 50 mg/kg body weight [21]. The experimental design comprised six groups: control (C), *Bifidobacterium bifidum*-only (B), diabetic (D), diabetic treated with *Bifidobacterium bifidum* (D+B), diabetic treated with metformin (D+M), and diabetic treated with the combination of *Bifidobacterium bifidum* and metformin (D+B+M). Each group included 10 animals (*n* = 10). Body weight was recorded weekly using a digital scale (Model: DS-200, Ohaus, Parsippany, NJ, USA) [22] throughout the experimental period. Fasting blood glucose (FBG) levels were determined after an overnight fast by collecting blood samples via tail vein puncture, with glucose concentrations measured using an enzymatic assay kit (Model: GLU-001, Roche Diagnostics, Rotkreutz, Switzerland) on an automated biochemical analyzer (COBAS Integra 400, Roche Diagnostics, Rotkreutz, Switzerland) [23]. All measurements were performed in triplicate to ensure reproducibility. Data were statistically analyzed using SPSS software (version 25, IBM, Armonk, NY, USA), employing one-way ANOVA followed by Tukey’s post hoc test, with significance set at *p* < 0.05.

#### 2.4.2. Glucose Tolerance and Insulin Sensitivity

Following the treatment period, animals were fasted overnight (12 h) with free access to water, and blood samples were collected via the tail vein for measurement of fasting blood glucose and insulin levels. Glucose concentrations were determined using the enzymatic assay kit (Model: GLU-001, Roche Diagnostics, Rotkreutz, Switzerland) [24], and insulin levels were quantified using an ELISA kit (Model: INS-101, Sigma-Aldrich, St. Louis, MO, USA) according to the manufacturers’ protocols [25]. HOMA-IR and HOMA-β were calculated using standard formulas (HOMA-IR = [fasting insulin (μU/mL) × fasting glucose (mmol/L)]/22.5; HOMA-β = [20 × fasting insulin (μU/mL)]/[fasting glucose (mmol/L) − 3.5]). The oral glucose tolerance test (OGTT) was conducted by administering a glucose load of 2 g/kg body weight orally, with blood samples collected at 0, 30, 60, 90, and 120 min post-administration. The area under the curve (AUC) was calculated to assess glucose clearance. The insulin sensitivity test (IST) involved a subcutaneous injection of insulin (0.75 U/kg) after a 6 h fast, with blood glucose measurements taken at 0, 15, 30, 45, and 60 min post-injection to evaluate peripheral glucose uptake. All assays were conducted in triplicate. Statistical analyses included one-way ANOVA with Tukey’s post hoc test (*p* < 0.05).

#### 2.4.3. Antioxidant Enzyme Levels

After the treatment period, liver tissues were rapidly excised from all experimental groups, rinsed in ice-cold saline, and homogenized in ice-cold phosphate-buffered saline (PBS) containing protease inhibitors. The homogenate was centrifuged at 10,000× *g* rpm for 15 min at 4 °C, and the supernatant was collected for enzyme assays. Catalase (CAT) activity was determined spectrophotometrically by monitoring the decomposition of hydrogen peroxide using a reagent kit (Model: CAT-001, Sigma-Aldrich, St. Louis, MO, USA) [12]. Superoxide dismutase (SOD) activity was measured by assessing the inhibition of nitroblue tetrazolium (NBT) reduction with an assay kit (Model: SOD-101, Roche Diagnostics, Rotkreutz, Switzerland) [26]. Glutathione peroxidase (GPx) activity was evaluated using a coupled assay with glutathione reductase using a kit (Model: GPx-202, Sigma-Aldrich, St. Louis, MO, USA) [27], and glutathione S-transferase (GST) activity was quantified by monitoring the conjugation of 1-chloro-2,4-dinitrobenzene (CDNB) with reduced glutathione using a kit (Model: GST-303, Roche Diagnostics, Rotkreutz, Switzerland) [28]. Protein concentrations were determined using the Bradford method (Model: Bradford-1000, Bio-Rad, Hercules, CA, USA) to normalize enzyme activities as units per mg protein [29]. Data were analyzed using SPSS software (version 25, IBM, Armonk, NY, USA), with statistical significance set at *p* < 0.05.

#### 2.4.4. Serum Lipid Profiles

Blood samples for serum lipid profile analysis were collected from rats after an overnight fast. Samples were allowed to clot at room temperature and then centrifuged at 3000 rpm for 15 min to separate the serum, which was stored at −80 °C until analysis. Serum lipid profiles, including low-density lipoprotein (LDL), free fatty acids (FFAs), triglycerides (TGs), total cholesterol (TC), and high-density lipoprotein (HDL), were measured using enzymatic assay kits (Model: X-123 for LDL, FFA, TG, and TC; Model: H-101 for HDL; Roche Diagnostics, Rotkreutz, Switzerland) on the COBAS Integra 400 biochemical analyzer [30]. All measurements were performed in triplicate, and data were analyzed using one-way ANOVA followed by Tukey’s post hoc test, with significance set at *p* < 0.05.

### 2.5. Histopathological Analysis of Liver Tissue

Liver samples were collected from all experimental groups following the treatment period and immediately immersed in 10% formalin solution (Model: F-10, Sigma-Aldrich, St. Louis, MO, USA) [31]. Fixed tissues were dehydrated gradually and embedded in paraffin (Model: P-20, Merck, Darmstadt, Germany) [32]. Sections of 4–5 µm thickness were cut using a Leica RM2255 microtome (Leica Biosystems, Nußloch, Germany) [33]. Staining with hematoxylin and eosin (H&E) and Masson’s trichrome was performed to evaluate general tissue architecture and fibrous tissue deposition. Slides were examined under a light microscope (Olympus BX51, Olympus Corporation, Tokyo, Japan) [34]. Statistical analysis of qualitative and quantitative histopathological data was performed using SPSS software (version 25, IBM, Armonk, NY, USA), with *p* < 0.05 considered statistically significant. All procedures adhered to ethical guidelines approved by the Institutional Animal Care and Use Committee (IACUC) [35].

### 2.6. Microbiome Analysis—DNA Extraction and Processing

Fecal DNA was extracted using the Qiagen QIAamp DNA Stool Mini Kit (Qiagen, Hilden, Germany) according to the manufacturer’s protocols, with an additional enzymatic digestion step included to minimize host DNA contamination. DNA purity and concentration were evaluated using a NanoDrop 2000 (Thermo Fisher Scientific, Waltham, MA, USA) at A260/A280 and A260/A230 ratios (acceptable range: 1.8–2.0).

Sequence quality was assessed using FastQC, and low-quality reads (Q < 20) were filtered out prior to downstream analysis. Rarefaction analysis was performed to normalize sequencing depth across samples to prevent bias in diversity comparisons.

Chimera filtering was conducted using the DADA2 algorithm within QIIME2, removing sequences < 100 base pairs. For taxonomic classification, raw sequencing reads were quality-filtered using QIIME2 pipelines. This included the removal of chimeric sequences, elimination of low-quality reads, and assignment of operational taxonomic units (OTUs) using the SILVA 138.1 reference database with a similarity threshold of 97%. Functional predictions of microbial communities were made using PICRUSt2 to infer metabolic pathways affected by *Bifidobacterium bifidum* treatment [36].

### 2.7. Statistical Analysis

All statistical analyses were conducted using R software (version 4.3.1, released June 2023) and GraphPad Prism (version 10.12, released 2023). Outlier detection was performed using Tukey’s method, and outliers were excluded before further analysis. For multiple comparisons, *p*-values were adjusted using the Benjamini–Hochberg false discovery rate (FDR) correction to control Type I errors. Data normality was assessed using the Shapiro–Wilk test before applying parametric tests. ANOVA was used for alpha diversity comparisons; if data were non-normally distributed, a Kruskal–Wallis test was applied instead. Kolmogorov–Smirnov and Levene’s tests were used to verify normality and homogeneity of variances, respectively. Beta diversity analysis was conducted using weighted and unweighted UniFrac distances, with group differences assessed using PERMANOVA (Permutational Multivariate Analysis of Variance, 999 permutations). Statistical significance was set at *p* < 0.05, and effect sizes were reported where applicable [37].

## 3. Results

### 3.1. Effect of Bifidobacterium bifidum on Body Weight and Fasting Blood Glucose Levels in Diabetic Rats Body Weight

As shown in Figure 1A, streptozotocin (STZ)-induced diabetes in the diabetic group (D) led to a significant 17.8% reduction in body weight compared to the control group (C) (*p* < 0.001). Administration of *Bifidobacterium bifidum* (diabetic + *Bifidobacterium bifidum* group, D+B) partially mitigated this weight loss, showing a 9.6% increase in body weight relative to the untreated diabetic group (*p* < 0.05). The diabetic + metformin group (D+M) exhibited a 7.8% increase in body weight compared to the diabetic group (*p* < 0.05). The combination treatment group (diabetic + *Bifidobacterium bifidum* + metformin, D+B+M) demonstrated the most pronounced improvement, with a 14.2% increase in body weight relative to the diabetic group (*p* < 0.01).

### 3.2. Fasting Blood Glucose (FBG)

Figure 1B demonstrates that FBG levels were significantly elevated in the diabetic group (256.8 ± 8.9 mg/dL) compared to the control group (89.7 ± 5.2 mg/dL, *p* < 0.001). *Bifidobacterium bifidum* supplementation in the D+B group resulted in a 32.4% reduction in FBG levels (173.6 ± 6.7 mg/dL, *p* < 0.01) relative to the untreated diabetic group. The D+M group also showed a significant decrease, with a 28.7% reduction in FBG levels (183.1 ± 7.2 mg/dL, *p* < 0.01). The combination group (D+B+M) exhibited the greatest reduction in blood glucose levels, with a 41.5% decrease (150.3 ± 6.1 mg/dL, *p* < 0.001), making it the most effective intervention among all treatment groups.

### 3.3. Effect of Bifidobacterium bifidum and Metformin, Alone and in Combination, on Glucose Tolerance and Insulin Sensitivity in Diabetic Rats

To assess glucose homeostasis and insulin sensitivity, HOMA-IR, HOMA-β, OGTT, and IST were evaluated (Figure 2). Diabetic rats exhibited a significant 2.8-fold increase in HOMA-IR compared to controls (7.5 ± 0.9 vs. 2.7 ± 0.3, *p* < 0.001), indicating severe insulin resistance. Treatment with metformin (D+M) reduced HOMA-IR by 42.3% (*p* < 0.01), while *Bifidobacterium bifidum* supplementation (D+B) resulted in a 48.6% reduction (*p* < 0.0001). The combination treatment (D+B+M) produced the most pronounced improvement, with a 65.2% reduction (*p* < 0.001).

Two-way ANOVA was performed to evaluate interaction effects between diabetes status and treatment type (DB, DM, and D+B+M). The analysis confirmed significant interactions, and pairwise comparisons using Tukey’s post hoc test revealed that the combination therapy (D+B+M) was significantly more effective than either the D+M or D+B groups (*p* < 0.05). Additionally, comparisons between the control (C) and probiotic-only (B) groups showed no statistically significant differences, confirming that *Bifidobacterium bifidum* does not alter glucose homeostasis in non-diabetic rats.

HOMA-β values, reflecting β-cell function, were significantly reduced in diabetic rats (31.2 ± 4.5% of control levels, *p* < 0.01). Metformin improved β-cell function by 55.0% (*p* < 0.05), *Bifidobacterium bifidum* by 63.4% (*p* < 0.01), and the combination treatment by 78.0% (*p* < 0.001). During the OGTT, diabetic rats showed delayed glucose clearance. Metformin reduced the glucose AUC by 25.0% (*p* < 0.05), *Bifidobacterium bifidum* by 27.9% (*p* < 0.01), and the combination treatment by 40.5% (*p* < 0.001), indicating improved glucose tolerance. The IST results demonstrated enhanced peripheral glucose uptake by 30.2% with metformin, 34.1% with *Bifidobacterium bifidum* (*p* < 0.05), and 45.3% with the combined treatment (*p* < 0.01) compared to the untreated diabetic group.

### 3.4. Effect of Bifidobacterium bifidum and Metformin on Antioxidant Enzyme Levels in Diabetic Rats

Figure 3 illustrates the effects of *Bifidobacterium bifidum* and metformin supplementation on antioxidant enzyme activities—catalase (CAT), superoxide dismutase (SOD), glutathione peroxidase (GPx), and glutathione S-transferase (GST)—in diabetic rats. The diabetic group (D) exhibited a significant reduction in CAT activity compared to the control (C) and probiotic-only (B) groups. In the diabetic + metformin (D+M) group, CAT activity was restored to approximately 60% of control levels (*p* < 0.05), while supplementation with *Bifidobacterium bifidum* in the diabetic + probiotic (D+B) group led to a 79.2% recovery of normal CAT levels (*p* < 0.05). The combination treatment (D+B+M) normalized CAT activity to roughly 88% of control levels (*p* < 0.01).

SOD activity was significantly decreased in the diabetic group compared to controls, with partial recovery observed in the D+M group (45% improvement, *p* < 0.05) and a 55.6% enhancement in the D+B group (*p* < 0.05). The D+B+M group exhibited the highest improvement, restoring SOD activity to nearly 70% of control levels (*p* < 0.01).

Similarly, GPx activity was markedly reduced in diabetic rats, with metformin and *Bifidobacterium bifidum* treatments increasing GPx activity by approximately 30% and 45%, respectively (*p* < 0.05). The combination treatment demonstrated the greatest enhancement of around 60% (*p* < 0.01).

Conversely, GST activity was significantly elevated in the diabetic group (a 45.2% increase over controls, *p* < 0.001). In the D+M group, GST activity was reduced by about 30% (*p* < 0.05), normalized in the D+B group (*p* < 0.05), and fully restored to control levels in the D+B+M group (*p* < 0.01).

### 3.5. Effects of Bifidobacterium bifidum on Serum Lipid Profiles in Diabetic Rats

Figure 4A–E illustrate the impact of *Bifidobacterium bifidum* supplementation, metformin treatment, and their combination on lipid metabolism in diabetic rats. Diabetes induction resulted in significant dyslipidemia, as evidenced by a 48.5% increase in low-density lipoprotein (LDL) (Figure 4A), 42.7% increase in free fatty acids (FFAs) (Figure 4B), 51.2% increase in triglycerides (TGs) (Figure 4C), and 38.9% increase in total cholesterol (TC) (Figure 4D) compared to the control group (C), while high-density lipoprotein (HDL) levels were reduced by 31.4% (Figure 4E) (*p* < 0.05).

In the diabetic group treated with metformin alone (D+M), modest improvements were observed, with LDL, FFA, TG, and TC levels reduced by approximately 15–20% and a moderate increase in HDL levels (Figure 4E) (*p* < 0.05). *Bifidobacterium bifidum* supplementation (D+B) yielded more pronounced effects, with reductions of 26.4% in LDL (Figure 4A), 22.5% in FFAs (Figure 4B), 28.3% in TGs (Figure 4C), and 21.8% in TC (Figure 4D), along with a 27.1% increase in HDL (Figure 4E) (*p* < 0.05). Notably, the combination treatment (D+B+M) produced the most marked benefits, reducing LDL, FFA, TG, and TC levels by 35.7%, 30.2%, 40.1%, and 32.5%, respectively (Figure 4A–D), and increasing HDL levels by 33.8% (Figure 4E) (*p* < 0.01).

These results highlight the synergistic lipid-lowering and cardioprotective effects of combining *Bifidobacterium bifidum* with metformin, underscoring their potential to restore lipid homeostasis and reduce cardiovascular risks under diabetic conditions.

As illustrated in Figure 4A–E, diabetes induction resulted in significant dyslipidemia, characterized by elevated levels of low-density lipoprotein (LDL) by 48.5%, free fatty acids (FFAs) by 42.7%, triglycerides (TGs) by 51.2%, and total cholesterol (TC) by 38.9%, alongside a 31.4% reduction in high-density lipoprotein (HDL) (*p* < 0.05). This lipid imbalance reflects the typical impaired lipid metabolism observed in diabetic conditions.

Supplementation with *Bifidobacterium bifidum* significantly improved the lipid profile, with reductions in LDL (−26.4%), FFA (−22.5%), TG (−28.3%), and TC (−21.8%) levels, accompanied by a 27.1% increase in HDL levels (*p* < 0.05). These improvements suggest that *B. bifidum* plays a critical role in restoring lipid homeostasis, potentially reducing cardiovascular risks associated with diabetes-induced dyslipidemia.

### 3.6. Histopathological Analysis of Liver Tissue Across Experimental Groups

Figure 5 presents the histopathological findings and quantitative assessments of liver tissue from experimental groups. Liver sections from the control group (Figure 5A) display normal hepatic architecture with well-preserved hepatocytes, intact sinusoids, and absence of inflammation or fibrosis, while sections from the *Bifidobacterium bifidum*-treated group (Figure 5B) are similar to the control. In contrast, the diabetic group (Figure 5C) shows marked hepatocyte degeneration, inflammatory cell infiltration, sinusoidal congestion, and fibrosis. Liver sections from diabetic rats treated with metformin (Figure 5D) exhibit moderate improvements with reduced inflammatory cell infiltration, less hepatocyte degeneration, and lower fibrosis levels. The diabetic group receiving *Bifidobacterium bifidum* (Figure 5E) shows reduced inflammation, improved hepatocyte morphology, and decreased fibrosis, and liver sections from the group treated with both *Bifidobacterium bifidum* and metformin (Figure 5F) display the most pronounced restoration of normal hepatic architecture. Quantitatively, the inflammation scores (Figure 5G) increased 4.3-fold in diabetic rats compared to controls (*p* < 0.001). In the diabetes + metformin group, inflammation was reduced by approximately 35% (*p* < 0.05), by 49.2% in the diabetes + *Bifidobacterium bifidum* group (*p* < 0.05), and by around 65% in the combined therapy group (*p* < 0.01). Hepatocyte degeneration (Figure 5H) was significantly elevated in diabetic rats (*p* < 0.001), with reductions of about 45% observed in the metformin-treated group (*p* < 0.05), 56.8% in the *Bifidobacterium bifidum*-treated group (*p* < 0.05), and 70% in the combined treatment group (*p* < 0.01). The fibrosis index (Figure 5I) increased 3.6-fold in diabetic rats (*p* < 0.001) and was reduced by roughly 35% with metformin (*p* < 0.05), 44.3% with *Bifidobacterium bifidum* (*Bifidobacterium bifidum*) supplementation (*p* < 0.05), and approximately 60% with the combined therapy (*p* < 0.01).

### 3.7. Microbiome Composition and Modulation by Bifidobacterium bifidum in Diabetic Rats

The gut microbiota composition varied significantly among the experimental groups (Figure 6). In the control group (C), *Firmicutes* constituted 78% and Bacteroidetes 12% of the microbiota, indicating a balanced microbial profile. In contrast, diabetic rats (D) exhibited a reduction in *Firmicutes* to 35% and an increase in Bacteroidetes to 45%, demonstrating marked dysbiosis. *Bifidobacterium bifidum* supplementation in diabetic rats (D+B) partially restored microbial balance, with *Firmicutes* increasing to 58% and Bacteroidetes decreasing to 32%. Diabetic rats treated with metformin (D+M) showed moderate improvements in microbiota composition, although these changes were less pronounced than those in the D+B group.

Microbial diversity, as measured using the Shannon diversity index (Figure 7), was highest in the control group (4.6 ± 0.12) and significantly declined in diabetic rats (2.4 ± 0.28). Treatment with *Bifidobacterium bifidum* in the D+B group improved diversity to 3.9 ± 0.18, while the D+M group also demonstrated an increase. The Firmicutes/Bacteroidetes (F/B) ratio (Figure 8) was reduced to 0.9 ± 0.15 in diabetic rats compared to 2.1 ± 0.1 in controls. *Bifidobacterium bifidum* administration in the D+B group improved the F/B ratio to 1.7 ± 0.14, and the D+M group exhibited intermediate values.

Heatmap analysis (Figure 9) confirmed the reduced abundance of Firmicutes and the elevated presence of Bacteroidetes in diabetic rats, with partial restoration observed following *Bifidobacterium bifidum* and metformin treatments. Additional analysis (Figure 10) of bacterial phyla proportions revealed similar trends, and a focused comparison (Figure 11) of Firmicutes versus Bacteroidetes further underscored the dysbiotic state in diabetic rats and the partial correction by the interventions.

Principal coordinate analysis (PCoA, Figure 12) demonstrated distinct clustering among the groups. The control (C) and *Bifidobacterium bifidum*-only (B) groups formed compact clusters, whereas the diabetic group (D) exhibited a dispersed pattern. The microbial profiles of the D+B and D+M groups shifted toward the control cluster, with the combined therapy (D+B+M) showing the most pronounced shift.

Principal coordinate analysis (PCoA) plot of microbial communities showing distinct clustering among the six experimental groups: control (C), *Bifidobacterium bifidum*-only (B), diabetic (D), diabetic + *Bifidobacterium bifidum* (D+B), diabetic + metformin (D+M), and diabetic + *Bifidobacterium bifidum* + metformin (D+B+M). Each sample within a group (e.g., C1, C2) is represented by a unique marker and color for clear differentiation. The control (C) and *Bifidobacterium bifidum*-only (B) groups exhibit tighter clustering, indicating a more stable and homogeneous microbiome, whereas the diabetic group (D) is dispersed, reflecting significant dysbiosis. Notably, the D+B and D+M groups shift closer to the control cluster, suggesting partial restoration of microbial stability, while the combined therapy group (D+B+M) shows the most pronounced move toward the control, indicating a stronger recovery of gut microbiota composition.

## 4. Discussion

The present study highlights the therapeutic potential of *Bifidobacterium bifidum* in mitigating diabetes-induced liver injury through modulating gut microbiota composition, enhancing hepatic function, and reducing oxidative stress. While previous studies have established correlations between probiotics and metabolic health, this study advances the field by elucidating specific mechanistic pathways underlying these beneficial effects.

The hepatoprotective properties of *Bifidobacterium bifidum* can be attributed to its ability to mitigate gut-derived endotoxin accumulation, particularly lipopolysaccharides (LPSs), which are known activators of the NF-κB signaling pathway. The reduction in LPS levels observed in this study likely disrupted this inflammatory cascade, leading to decreased oxidative stress and hepatic fibrosis. This aligns with existing literature on probiotic-mediated inflammation modulation but introduces novel insights into how specific gut microbiota alterations influence hepatic inflammation [38,39].

Additionally, improved glycemic control observed with *Bifidobacterium bifidum* supplementation may be linked to increased production of short-chain fatty acids (SCFAs), particularly butyrate, which enhances insulin sensitivity via G-protein-coupled receptors (GPR41 and GPR43) on pancreatic β-cells. This study corroborates previous findings on probiotics and glycemic regulation [40,41], while offering a mechanistic explanation for the observed reduction in homeostasis model assessment for insulin resistance (HOMA-IR) and enhancement of β-cell function (HOMA-β).

A critical aspect of this study is the restoration of gut microbiota balance, a hallmark of metabolic dysregulation in diabetes. Diabetic rats exhibited a dysbiotic gut microbiome characterized by a reduced Firmicutes/Bacteroidetes (F/B) ratio and diminished microbial diversity, which are well-documented features of type 2 diabetes [42,43]. *Bifidobacterium bifidum* supplementation reversed these imbalances by promoting beneficial *Firmicutes* populations and suppressing opportunistic pathogens. Unlike previous studies that broadly associate probiotics with gut health improvements, this research provides specific mechanistic insights into how microbial shifts contribute to enhanced gut barrier integrity and metabolic resilience [44,45,46].

Oxidative stress and chronic inflammation play pivotal roles in diabetes-induced hepatic injury. The enhancement of antioxidant defense mechanisms, evidenced by increased activities of superoxide dismutase (SOD), catalase (CAT), and glutathione peroxidase (GPx), suggests an activation of the Nrf2 signaling pathway, which regulates antioxidant response elements (AREs) [47,48]. Concurrent reductions in hepatic fibrosis and inflammatory markers further support a dual regulatory role, where *Bifidobacterium bifidum* suppresses NF-κB-mediated inflammation while activating cytoprotective antioxidant pathways.

Furthermore, this study demonstrates improvements in lipid metabolism, characterized by decreased levels of low-density lipoprotein (LDL), triglycerides (TGs), and total cholesterol (TC), alongside increased high-density lipoprotein (HDL). These effects may stem from *Bifidobacterium bifidum*’s influence on bile acid metabolism and fatty acid oxidation. Mechanistically, this aligns with evidence suggesting that probiotics modulate cholesterol metabolism via bile salt hydrolase (BSH) activity and activation of AMP-activated protein kinase (AMPK) pathways [15,49]. These findings contribute to a more comprehensive understanding of how probiotics affect lipid metabolism in metabolic disorders.

## 5. Significance and Future Research Perspectives

This study provides a mechanistic framework for understanding how *Bifidobacterium bifidum* exerts protective effects against diabetes-induced liver injury, reinforcing its potential as a microbiome-targeted therapeutic intervention. However, several key areas require further exploration:1.Clinical Translation and Long-Term Effects
oWhile preclinical findings are promising, controlled clinical trials in diabetic patients are essential to validate the therapeutic potential of *Bifidobacterium bifidum* and establish standardized dosing regimens. Future studies should assess its long-term effects on hepatic and metabolic outcomes in diverse patient populations.
2.Microbiome-Driven Therapeutic Strategies
oInvestigating the synergistic effects of *Bifidobacterium bifidum* with other probiotic strains or dietary interventions could enhance its efficacy. Multi-omics approaches, including metagenomics and metabolomics, should be employed to map host–microbiome interactions at a systems level.
3.Synbiotic and Postbiotic Applications
oThe development of synbiotic formulations combining *Bifidobacterium bifidum* with prebiotics that selectively promote its growth may optimize its therapeutic benefits. Additionally, exploring the potential of postbiotics—bioactive compounds derived from probiotics—could offer novel treatment strategies.
4.Comparative Efficacy with Conventional Therapies
oFuture studies should compare *Bifidobacterium bifidum* supplementation with conventional anti-diabetic and hepatoprotective treatments to determine its relative efficacy and potential for integration into current treatment protocols.
5.Mechanistic Elucidation Using Advanced Models
oEmploying humanized gut microbiota models and in vitro hepatic organoid systems may provide deeper mechanistic insights into the host–microbe interactions driving *Bifidobacterium bifidum*’s protective effects.

By addressing these research gaps, future studies could pave the way for microbiome-based therapeutic interventions that not only mitigate diabetes-induced liver injury but also contribute to broader metabolic disease management.

## 6. Conclusions

The results of this study demonstrate that *Bifidobacterium bifidum* supplementation exerts significant protective effects on liver function and metabolic parameters in diabetic rats with hepatic damage. By restoring gut microbiota balance, enhancing glycemic control, and reducing oxidative stress, *Bifidobacterium bifidum* shows substantial potential for alleviating liver complications associated with diabetes. These findings highlight the critical systemic influence of the gut microbiota on liver health in diabetes-related hepatic disease and reinforce the therapeutic potential of probiotic-based interventions. Further research is needed to optimize probiotic treatment strategies, evaluate their long-term efficacy, and explore their applicability in clinical settings to effectively safeguard liver function and improve metabolic health under diabetic conditions.

## Figures and Tables

**Figure 1 biology-14-00323-f001:**
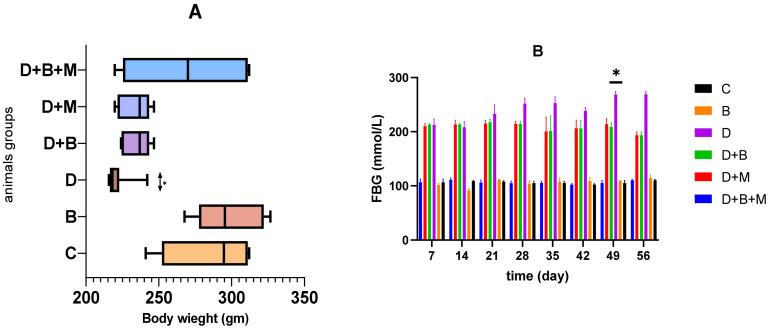
(**A**,**B**) *Bifidobacterium bifidum* and metformin improve body weight and glycemic control in diabetic rats. (**A**) Body weight changes across experimental groups during the study. The control group (C) maintained a stable weight, while the diabetic group (D) experienced significant weight loss following streptozotocin (STZ)-induced diabetes. *Bifidobacterium bifidum* treatment (D+B) mitigated this loss, showing a 9.6% increase in body weight compared to the untreated diabetic group (*p* < 0.05). Metformin treatment (D+M) also resulted in a moderate weight recovery, while the combination treatment (D+B+M) exhibited the most substantial weight gain, indicating a potential synergistic effect of probiotic and pharmacological interventions in restoring metabolic balance. (**B**) Fasting blood glucose (FBG) levels among experimental groups. Diabetic rats (D) showed significantly higher FBG levels compared to the control group (C), confirming STZ-induced hyperglycemia. Administration of *Bifidobacterium bifidum* (D+B) resulted in a 32.4% reduction in FBG (*p* < 0.01), similar to the effect observed in the metformin-treated group (D+M). The combination treatment (D+B+M) demonstrated the greatest reduction in blood glucose levels, suggesting enhanced efficacy of dual therapy in improving glycemic control. Data are presented as mean ± standard error (SE), with statistical significance determined using one-way ANOVA followed by Tukey’s post hoc test. Data are presented as mean ± standard deviation (*n* = 8–10), with statistical significance denoted as ** p* < 0.001.

**Figure 2 biology-14-00323-f002:**
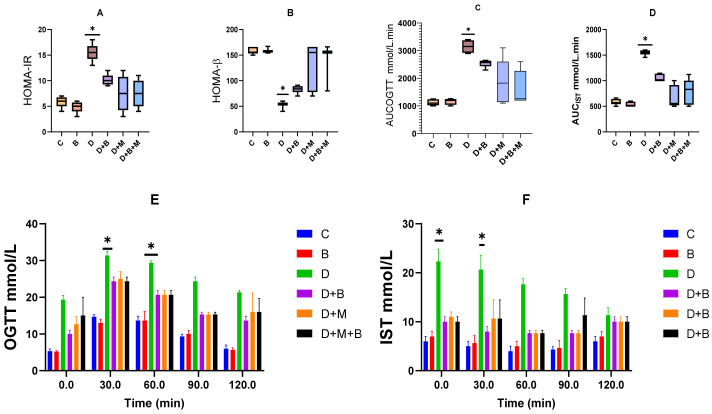
(**A**,**B**) *Bifidobacterium bifidum* and metformin enhance glucose tolerance and insulin sensitivity in diabetic rats. Homeostatic model assessment for insulin resistance (HOMA-IR) and beta-cell function (HOMA-β) in different experimental groups. Diabetic rats (D) exhibited a significant increase in HOMA-IR and a decline in HOMA-β compared to controls (C), indicating severe insulin resistance and β-cell dysfunction. Treatment with *Bifidobacterium bifidum* (D+B) and metformin (D+M) significantly improved these parameters, with the combination therapy (D+B+M) showing the most pronounced effect. (**C**,**D**) The area under the curve (AUC) for the oral glucose tolerance test (OGTT) and insulin sensitivity test (IST) demonstrates improved glucose clearance and peripheral insulin sensitivity in treated groups. *Bifidobacterium bifidum* and metformin, both individually and in combination, significantly reduced AUC values compared to the diabetic group, with the combination therapy yielding the greatest improvement. (**E**,**F**) OGTT and IST outcomes at week 8 further confirm enhanced glucose tolerance and insulin sensitivity in treated rats, particularly in the D+B+M group. Data are presented as mean ± standard deviation (*n* = 8–10), with statistical significance denoted as ** p* < 0.001.

**Figure 3 biology-14-00323-f003:**
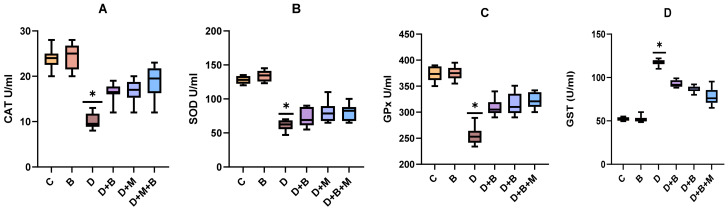
(**A**–**D**) *Bifidobacterium bifidum* and metformin enhance antioxidant enzyme activities in diabetic rats. The effects of *Bifidobacterium bifidum* and metformin on antioxidant enzyme activities, including catalase (CAT), superoxide dismutase (SOD), glutathione peroxidase (GPx), and glutathione S-transferase (GST), in diabetic rats. (**A**,**B**) CAT and SOD activities were significantly reduced in the diabetic group (D) compared to the control group (C). Treatment with *Bifidobacterium bifidum* (D+B) and metformin (D+M) significantly restored enzyme activity, with the combination therapy (D+B+M) showing the highest recovery to near-normal levels. (**C**) GPx activity, which was also diminished in diabetic rats, increased by 30% following metformin treatment and by 45% with *Bifidobacterium bifidum*, with the combination treatment leading to a 60% enhancement. (**D**) GST activity was significantly elevated in the diabetic group, but metformin treatment partially reduced it, *Bifidobacterium bifidum* normalized it, and the combination therapy fully restored GST activity to control levels. These findings suggest that probiotic and pharmacological interventions act synergistically to enhance antioxidant defenses against diabetes-induced oxidative stress. Data are presented as mean ± standard deviation (SD) (*n* = 10), with statistical significance denoted as *p* < 0.05, *p* < 0.01, and * *p* < 0.001 compared to the diabetic group.

**Figure 4 biology-14-00323-f004:**
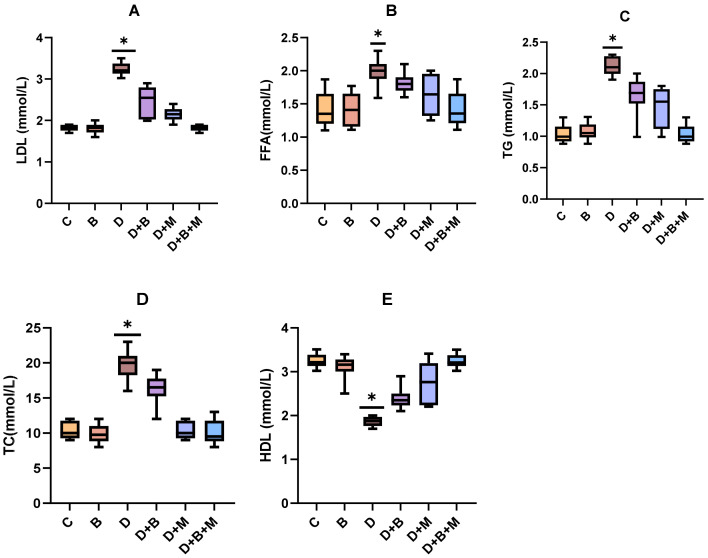
(**A**–**E**) *Bifidobacterium bifidum* and metformin modulate serum lipid profiles in diabetic rats. The effect of *Bifidobacterium bifidum*, metformin, and their combination on serum lipid profile parameters in diabetic rats. (**A**) Low-density lipoprotein (LDL) levels were significantly elevated in the diabetic group (D) compared to controls (C), while treatment with *Bifidobacterium bifidum* (D+B), metformin (D+M), and their combination (D+B+M) significantly reduced LDL levels, with the combination therapy showing the greatest reduction. (**B**) Free fatty acids (FFAs) were markedly increased in diabetic rats but were significantly lowered following probiotic and metformin treatments, particularly in the combination group. (**C**) Triglyceride (TG) levels followed a similar trend, with diabetic rats exhibiting a sharp increase that was significantly ameliorated by treatment, especially in the D+B+M group. (**D**) Total cholesterol (TC) was also elevated in diabetic rats but reduced following *Bifidobacterium bifidum* and metformin administration, with the greatest reduction observed in the combination therapy group. (**E**) High-density lipoprotein (HDL) levels, which were significantly decreased in diabetic rats, were restored by both *Bifidobacterium bifidum* and metformin, with the combination treatment demonstrating the highest HDL increase. Data are presented as mean ± standard deviation (*n* = 10), with statistical significance denoted as *p* < 0.05, *p* < 0.01, and * *p* < 0.001 compared to the diabetic group.

**Figure 5 biology-14-00323-f005:**
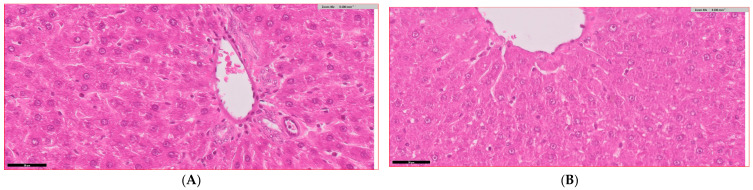
(**A**–**I**) *Bifidobacterium bifidum* and metformin protect against diabetes-induced liver injury. (**A**–**F**) Histopathological liver sections and (**G**–**I**) quantitative assessment of liver tissue across experimental groups demonstrate the protective effects of *Bifidobacterium bifidum* and Metformin—alone and in combination—against diabetes-induced liver injury. (**A**,**B**) Liver sections from the control (C) and *Bifidobacterium bifidum*-treated (B) groups exhibit normal hepatic architecture with preserved hepatocytes and intact sinusoids. (**C**) The diabetic group (D) shows severe hepatocyte degeneration, inflammatory infiltration, sinusoidal congestion, and fibrosis. (**D**) Treatment with metformin (D+M) leads to moderate improvement in liver histology. (**E**) The *Bifidobacterium bifidum*-treated group (D+B) demonstrates further histological enhancement, with reduced inflammation and improved hepatocyte integrity. (**F**) The combination treatment (D+B+M) achieves the most pronounced restoration of normal hepatic architecture. (**G**) The inflammation score increased 4.3-fold in diabetic rats (*p* < 0.001) but was reduced by approximately 35% with metformin, 49.2% with *Bifidobacterium bifidum*, and 65% with the combination therapy. (**H**) Hepatocyte degeneration, significantly elevated in diabetic rats (*p* < 0.001), was reduced by 45% with Metformin, 56.8% with *Bifidobacterium bifidum*, and 70% with the combination treatment. (**I**) Fibrosis levels, also significantly increased in diabetic rats, were reduced by 35%, 44.3%, and 60% following metformin, *Bifidobacterium bifidum*, and combination therapy, respectively. These findings highlight the hepatoprotective potential of probiotic and pharmacological interventions in mitigating diabetes-induced liver damage. Data are presented as mean ± standard deviation (*n* = 10), with statistical significance denoted as *p* < 0.05, *p* < 0.01, and * *p* < 0.001 compared to the diabetic group.

**Figure 6 biology-14-00323-f006:**
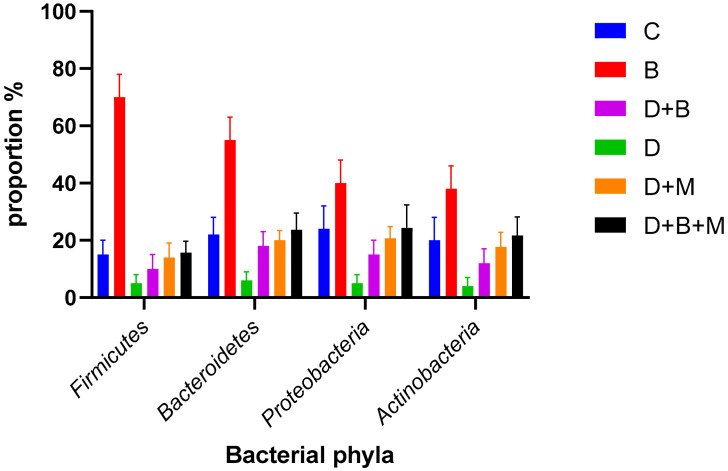
Bacterial phyla proportions across the six experimental groups. Bars represent the relative abundance (%) of *Firmicutes* (F), *Bacteroidetes* (B), *Proteobacteria* (P), and *Actinobacteria* (A) in the following groups: control (C), *Bifidobacterium bifidum*-only (B), diabetic + *Bifidobacterium bifidum* (D+B), Diabetic (D), diabetic + metformin (D+M), and diabetic + *Bifidobacterium bifidum* + metformin (D+B+M). Data are presented as mean ± standard error (*n* = 8–10). Note the substantial decrease in Firmicutes and increase in Bacteroidetes in the diabetic group (D) compared to the control (C). *Bifidobacterium bifidum* and/or metformin interventions partially restore microbial balance, with the greatest improvement observed in the combined therapy group (D+B+M).

**Figure 7 biology-14-00323-f007:**
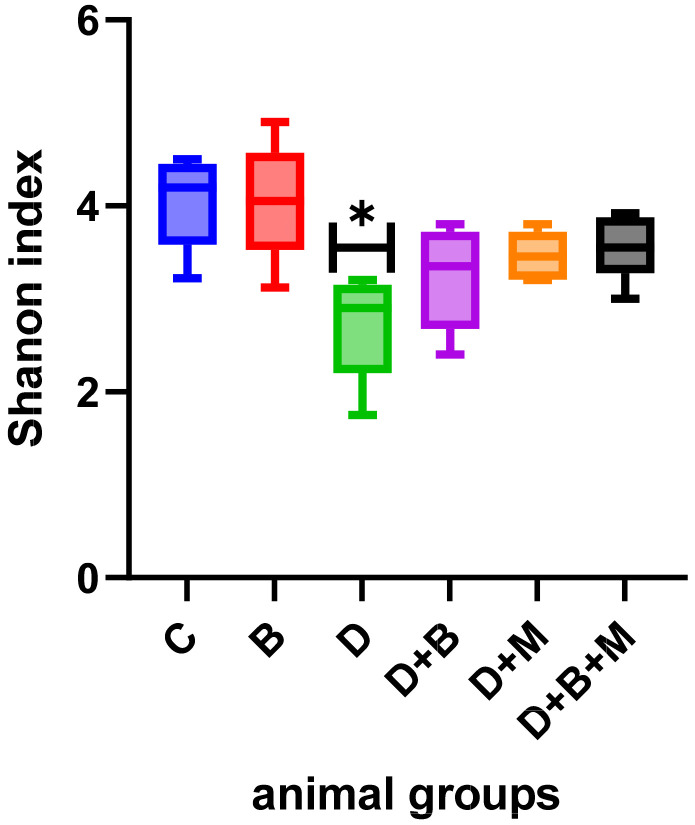
Shannon diversity index across the six experimental groups: control (C), *Bifidobacterium bifidum*-only (B), diabetic (D), diabetic + *Bifidobacterium bifidum* (D+B), diabetic + metformin (D+M), and diabetic + *Bifidobacterium bifidum* + metformin (D+B+M). The Shannon index measures microbial diversity, with higher values indicating a more diverse gut microbiota. The diabetic group (D) shows a significant decrease (* *p* < 0.05) compared to the control group (C). Both *Bifidobacterium bifidum* and metformin treatments partially restore microbial diversity, and the combination therapy (D+B+M) exhibits the most pronounced improvement. Data are presented as box-and-whisker plots (*n* = 8–10).

**Figure 8 biology-14-00323-f008:**
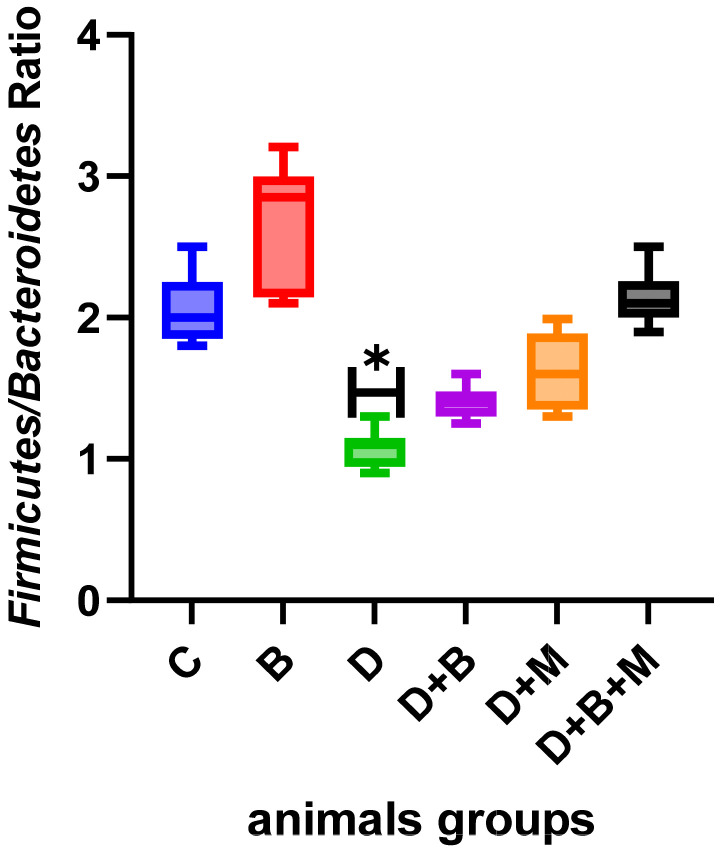
The Firmicutes/Bacteroidetes (F/B) ratio across the six experimental groups: control (C), *Bifidobacterium bifidum*-only (B), diabetic (D), diabetic + *Bifidobacterium bifidum* (D+B), diabetic + metformin (D+M), and diabetic + *Bifidobacterium bifidum* + metformin (D+B+M). The control group (C) exhibits the highest F/B ratio, reflecting a healthy microbiota composition. In contrast, the diabetic group (D) shows a marked reduction in the F/B ratio (* *p* < 0.05), indicative of gut dysbiosis. *Bifidobacterium bifidum* supplementation (D+B) and metformin treatment (D+M) each partially restore the F/B ratio, with the combination therapy (D+B+M) demonstrating a more pronounced improvement. Error bars represent the standard error of the mean (SEM), and statistical significance is indicated with an asterisk.

**Figure 9 biology-14-00323-f009:**
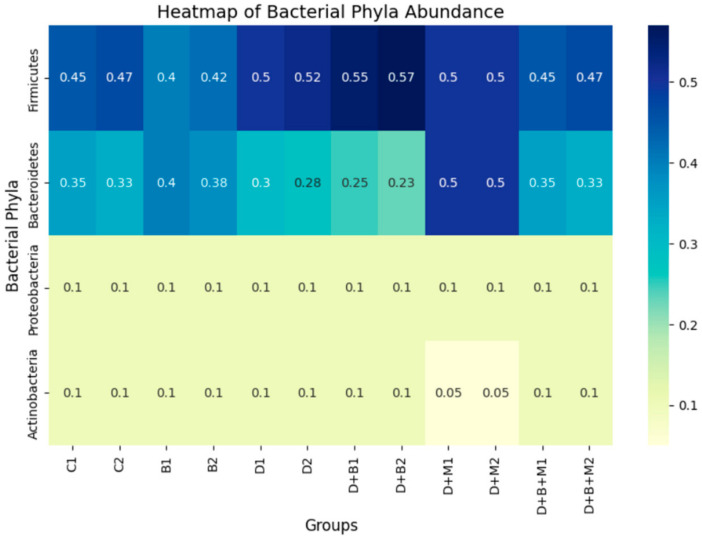
Bacterial abundance heatmap across the six experimental groups: control (C), *Bifidobacterium bifidum*-only (B), diabetic (D), diabetic + *Bifidobacterium bifidum* (D+B), diabetic + metformin (D+M), and diabetic + *Bifidobacterium bifidum* + metformin (D+B+M). Each group includes two biological replicates (e.g., C1, C2). The color gradient (from light/yellow to dark/blue) represents the relative abundance of the main bacterial phyla—Firmicutes, Bacteroidetes, Proteobacteria, and Actinobacteria—with lighter shades indicating lower proportions and darker shades indicating higher proportions. Numerical values within each cell denote the exact relative abundance. The control group (C) shows a predominance of Firmicutes, while the diabetic group (D) exhibits a marked reduction in Firmicutes and an increase in Bacteroidetes and other phyla, reflecting gut dysbiosis. *Bifidobacterium bifidum* supplementation (B, D+B) and metformin treatment (D+M) each influence microbial composition, whereas the combination therapy (D+B+M) demonstrates the most pronounced restoration of beneficial bacterial phyla.

**Figure 10 biology-14-00323-f010:**
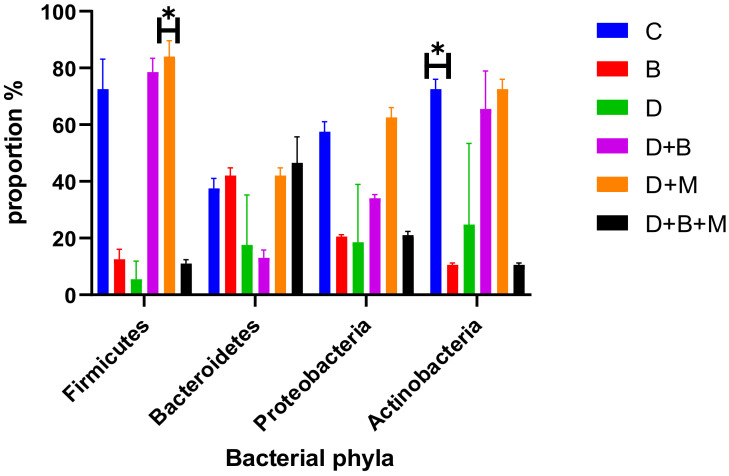
Proportions of the main bacterial phyla across the six experimental groups: control (C), *Bifidobacterium bifidum*-only (B), diabetic (D), diabetic + *Bifidobacterium bifidum* (D+B), diabetic + metformin (D+M), and diabetic + *Bifidobacterium bifidum* + metformin (D+B+M). Bars represent the mean relative abundance (%) of *Firmicutes*, *Bacteroidetes*, *Proteobacteria*, and *Actinobacteria* (*n* = 8–10 per group), with error bars indicating the standard deviation. The diabetic group (D) shows a marked reduction in Firmicutes and an increase in Bacteroidetes compared to the control group (C), indicative of dysbiosis. *Bifidobacterium bifidum* supplementation (B, D+B) and metformin treatment (D+M) each improve microbial composition, while the combination therapy (D+B+M) demonstrates the most pronounced restoration of phyla balance. Asterisks (*) denote statistically significant differences compared to the diabetic group (*p* < 0.05).

**Figure 11 biology-14-00323-f011:**
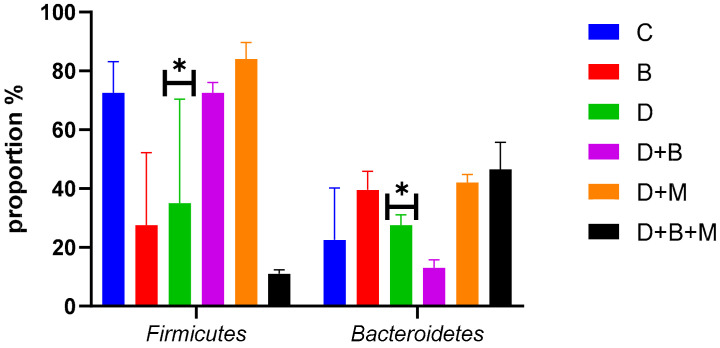
The *Firmicutes* vs. *Bacteroidetes* proportions in the six experimental groups: control (C), *Bifidobacterium bifidum*-only (B), diabetic (D), diabetic + *Bifidobacterium bifidum* (D+B), diabetic + metformin (D+M), and diabetic + *Bifidobacterium bifidum* + metformin (D+B+M). Bars represent the mean relative abundance (%) ± standard deviation (*n* = 8–10). The diabetic group (D) shows a notable reduction in Firmicutes and an increase in Bacteroidetes compared to the control (C), reflecting dysbiosis. *Bifidobacterium bifidum* supplementation (B, D+B) and metformin treatment (D+M) each improve the *Firmicutes/Bacteroidetes* ratio, while the combination therapy (D+B+M) demonstrates the most pronounced restoration of microbial balance. Asterisks (*) indicate significant differences compared to the diabetic group (*p* < 0.05).

**Figure 12 biology-14-00323-f012:**
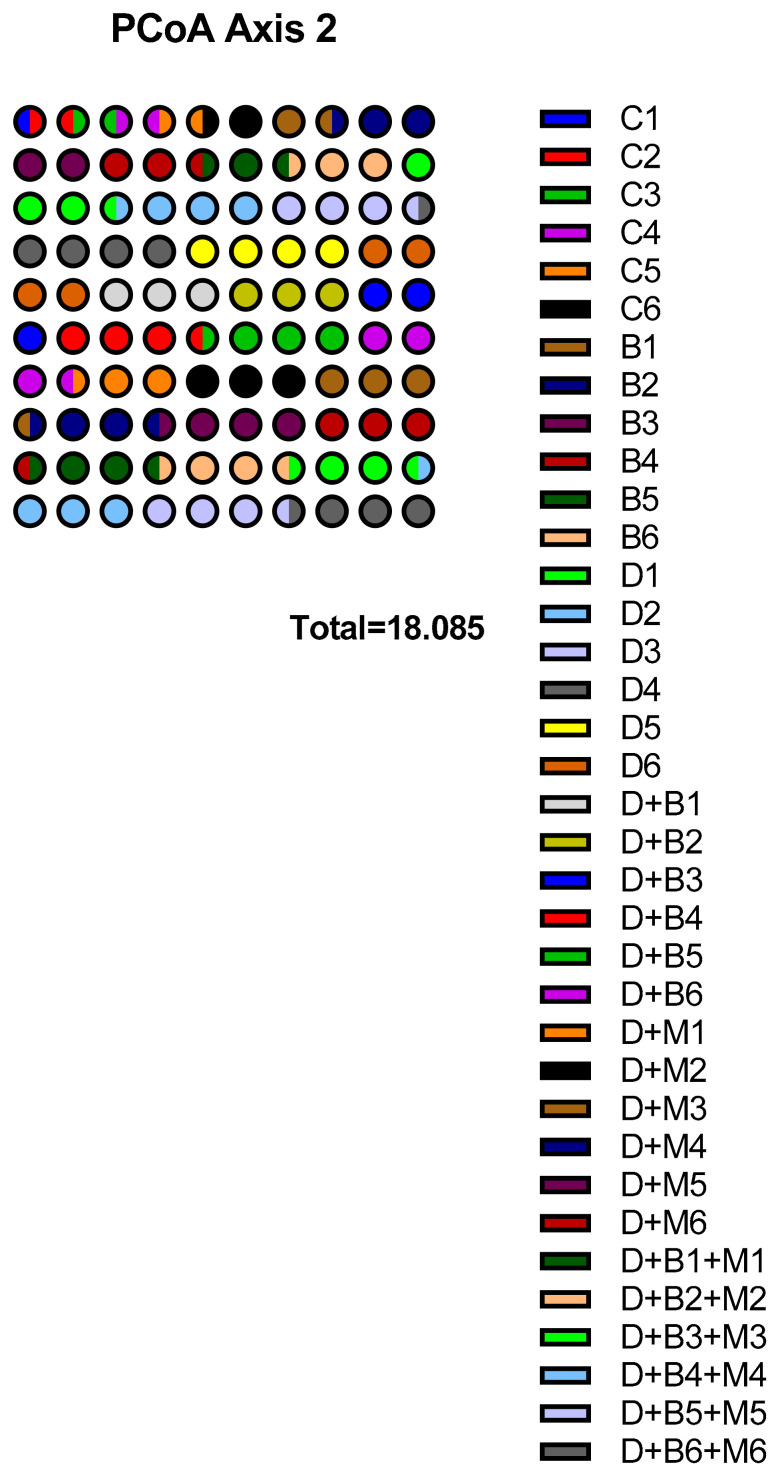
Principal coordinate analysis (PCoA) of microbial communities across experimental groups.

## Data Availability

All datasets generated or analyzed during this study are included in the manuscript.

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
